# A redox-neutral synthesis of ketones by coupling of alkenes and amides

Jing Li[1,3], Rik Oost[1,3], Boris Maryasin[1,2], Leticia González [2] & Nuno Maulide [1]

The direct synthesis of ketones via carbon–carbon bond formation represents one of the most important challenges in organic synthesis. Hydroacylation of alkenes offers perhaps the most efficient and atom-economical approach for the preparation of ketones employing carbonyl compounds and alkenes as feedstocks. State-of-the-art hydroacylation is typically achieved by a transition metal-catalysed coupling of an aldehyde and an alkene but is plagued by competing decarbonylation, requiring the installation of directing groups in the aldehyde reactant. Herein, we present a method for the hydroacylation of alkenes employing amides in a metal-free regime, proceeding by a new mechanism and offering orthogonal reactivity to the conventional, metal-catalysed alternatives.

[1] Institute of Organic Chemistry, University of Vienna, Währinger Strasse 38, 1090 Vienna, Austria. [2] Institute of Theoretical Chemistry, University of Vienna, Währinger Strasse 17, 1090 Vienna, Austria. [3]These authors contributed equally: Jing Li, Rik Oost. Correspondence and requests for materials should be addressed to N.M. (email: nuno.maulide@univie.ac.at)

**K**etones and aldehydes are perhaps the quintessential functional groups of organic chemistry. Their unique ability to mediate C–C bond forming reactions serving as either electrophiles (by direct nucleophilic addition to the carbonyl) or nucleophiles (by virtue of enolate or enamine formation) remains one of the cornerstones of the past 4 decades of organic synthesis. An overabundance of ketone syntheses rely on the direct, one-step 1,2-addition of organometallic reagents to suitable electrophilic carboxylic acid derivatives[1,2]. Unfortunately, these nucleophilic substitutions can suffer from a number of limitations including overaddition, poor chemoselectivity regarding the presence of other carbonyl groups, excessive use of acylating reagents or tedious procedures. The advent of the venerable Weinreb amides and related derivatives[3,4] offered a robust solution to the overaddition issue and triggered the development of related procedures relying on amide activation/organometallic addition (Fig. 1a)[5]. The recent breakthrough of Garg, Szostak, and others on achieving ketone synthesis from activated amides via Pd-catalysed[6] or Ni-catalysed[7–9] cross-coupling is perhaps the corollary of these developments (Fig. 1b). Nevertheless, both the Weinreb family of reactions and these elegant cross-coupling processes still rely on stoichiometric amounts of main group organometallics or organoboron reagents.

Alkenes would constitute highly appealing alternatives to conventional organometallic reagents for addition to carbonyl groups en route to ketone synthesis[10–12]. Hydroacylation of olefins has therefore emerged as a robust method for the preparation of ketones, typically achieved by the atom-economical, catalytic addition of an aldehyde C−H bond across an alkene[13,14]. However, contemporary transition metal-catalysed hydroacylation presents several challenges, the most notable of which is the competing decarbonylation of the aldehyde partner prior to coupling with the alkene counterpart (Fig. 1c). Therefore, while intramolecular olefin hydroacylation has reached high levels of efficiency and selectivity, intermolecular hydroacylation still typically relies on (a) aldehydes equipped with directing groups designed to minimize decarbonylation as well as on (b) activated alkene partners.

Aiming to address these challenges and guided by previous work on amide activation[5,15–19], we herein report an approach to olefin hydroacylation relying on an intermolecular coupling of secondary amides and alkenes (Fig. 1d), which requires neither a transition metal catalyst nor directing groups and which delivers ketones with high levels of chemo- and regioselectivity.

## Results

**Reaction discovery and optimization.** In initial experiments, a range of secondary amides **1** was activated prior to the addition of commercially available alkene **2a** (Table 1). From the outset, trace amounts of ketone could be detected using the simple *N*-propyl amide **1a** (Table 1, entry 1). Encouraged by this result, we noted that the amide *N*-substituent critically affects the efficiency of this transformation (entries 1–6). Eventually, we found that *N*-allylamides afforded high (80–90%) and reproducible yields of the ketone **3a**, a formal hydroacylation product. Additional screening of base (Supplementary Table 1) and solvent effects revealed that CH$_2$Cl$_2$ and CH$_3$CN afforded the desired ketone product in similar yields. (cf. entry 6 and entry 7).

**Substrate scope.** With optimized conditions in hand, we assessed the scope of the methodology (Table 2). The functional group tolerance and substrate scope on the amide component of this transformation were investigated first. Different aromatic amides bearing valuable substituents for downstream reactions (such as –CN, –NO$_2$, –COOMe, –MeCO, –CHO, –Cl and –Br) were tolerated. Functional groups such as a ketone **3e**, an aldehyde **3f** or a boronic ester **3i** also tolerated in moderate to good yields[20]. In addition, the orthogonality of this process in respect to

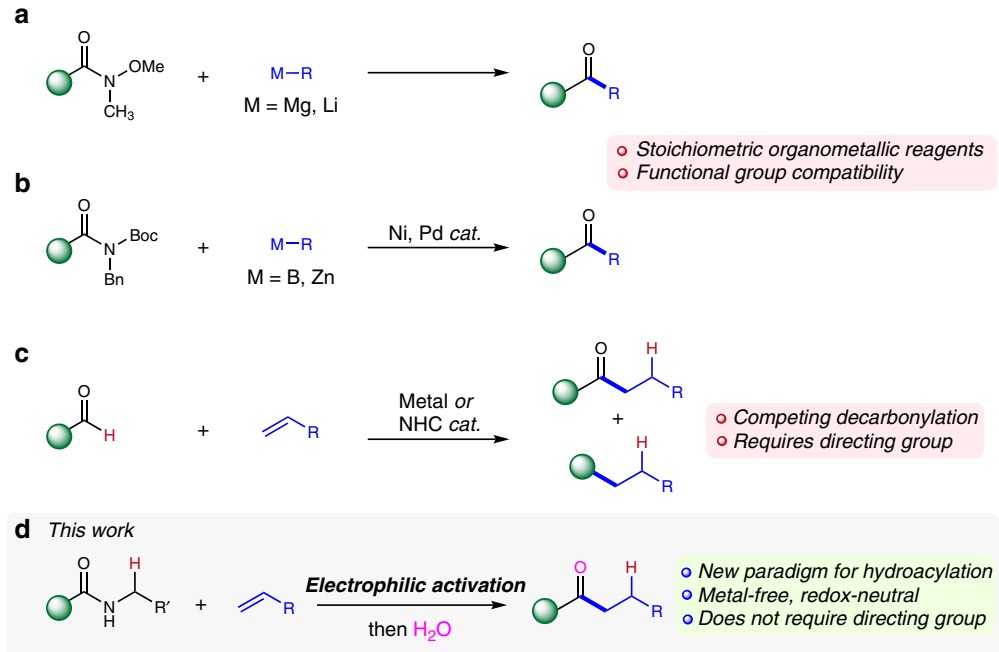

**Fig. 1** Examples for ketone synthesis in modern organic chemistry and hydroacylation method using secondary amides. **a** 1,2-monoaddition of organometallic reagents to Weinreb amides. **b** Pd- and Ni-catalysed cross-coupling of activated amides. **c** Challenges in contemporary olefin hydroacylation. **d** The present study: Intermolecular hydroacylation of olefins and amides

**Table 1 Reaction discovery and optimization[a]**

| Entry | R | | Yield (%) |
|-------|---|---|-----------|
| 1 | Me | 1a | 12 |
| 2 | Me, Me | 1b | 50 |
| 3 | cyclopentyl | 1c | 40 |
| 4 | cyclohexyl | 1d | 50 |
| 5 | benzyl | 1e | <5 |
| 6 | allyl | 1f | 86 |
| 7[b] | allyl | 1f | 81 |

*DCM* dichloromethane, *CH₃CN* acetonitrile, *Tf₂O* trifluoromethanesulfonic anhydride, *2-F-Py* 2-fluoropyridine
[a]Amide **1** (0.2 mmol), Tf₂O (0.22 mmol), 2-F-Py (0.22 mmol), alkene **2a** (0.4 mmol) in DCM (0.1 M) at 0 °C for 2 h and warmed up to room temperature for 14 h under argon; isolated yield after chromatography
[b]CH₃CN instead of CH₂Cl₂

hydroacylation of simple linear alpha alkenes/alkynes becomes apparent when examining aliphatic amides **3u–3v**, whereby alkene and alkyne moieties entailed in the amide partner behaved as spectators in this process.

At this juncture, we turned our attention to the scope of alkenes. Different α-substituted styrenes were tested and it was observed that steric hindrance does not affect reactivity in a pronounced manner (**4a–4e**). Electron-withdrawing substituents such as –CF₃ or –CN (**4j**, **4k**) still afforded the desired ketone in good chemical yield. More importantly, a simple 1,3-diene (**4f**) was also a suitable substrate for this transformation. The use of an allyl silane led to what could be termed an "interrupted allylation", delivering a product (**4**) where silicon is retained. The methodology was also applied with success to heterocyclic and more complex amides with a drug-like framework[21,22] containing Lewis basic pyrimidine moieties (**5a**) or the late-stage functionalization of dehydrocholic acid (**5b**).

**Mechanistic studies**. We then sought to obtain experimental information on the reaction mechanism (Fig. 2). The use of deuterated amide **d₁-1b** led to selective deuterium incorporation at the carbon β-to the ketone carbonyl (Fig. 2a). This is suggestive of a hydride transfer event. On the other hand, the use of bisdeuterated α-methylstyrene **d₂-2a** led to deuterium incorporation in the α-position of ketone **d₂-3b** (Fig. 2b). Finally, quenching the reaction with H₂¹⁸O (20 equiv.) generated a ketone product whereby the incorporation of ¹⁸O is > 90% (Fig. 2c). These observations allowed us to put forth a mechanistic proposal as depicted in Fig. 2d. Thus, activation of the amide starting material is likely to generate an *N*-allyl

nitrilium species **6** (Supplementary Fig. 4)[23], for which we also have obtained in situ NMR evidence (Supplementary Fig. 5). This electrophile is captured by the alkene partner, setting up the stage for intramolecular 1,5-hydride delivery. This step simultaneously achieves reduction of the transient carbocation **7** and formation of an azoniaallene intermediate **8**, the hydrolysis of which results in the hydroacylation products. Upon hydrolysis, ammonia and an aldehyde are formally excised from **8**[24].

**Computational studies**. We have performed quantum chemical calculations in order to get a deeper insight into the reaction mechanism. Figure 3 shows the computed reaction profile for formation of the final intermediate **8** (Fig. 3, right) starting from the nitrilium intermediate **6**. The same figure compares this to the side-reaction of nitrilium fragmentation, yielding benzonitrile (Fig. 3, left). For the state **6'** in Fig. 3 the alkene and nitrilium species are taken separately, while **6** (used as a reference 0.0 kcal mol⁻¹) is the reactant complex before the transition state **TS₆₋₇**.

The first step of the main reaction (Fig. 3, right) is formation of the intermediate **7** via transition state **TS₆₋₇**. Noteworthy, the triflate anion is found to be covalently bound to the originally postulated carbenium ion. The newly formed C-O bond undergoes cleavage concomitantly with 1,5-hydride transfer in the following step, leading to the intermediate **8** via transition state **TS₇₋₈**. Intermediate **7** contains a double C-N bond and can, therefore, exist in both Z and E forms. Possible diastereomeric pathways, resulting from the presence in intermediate **8** of two stereogenic elements, were also considered (Supplementary Fig. 13). The fragmentation side-reaction[25] (Fig. 3, left) proceeds

**Table 2 Scope of the metal-free hydroacylation of alkenes[a]**

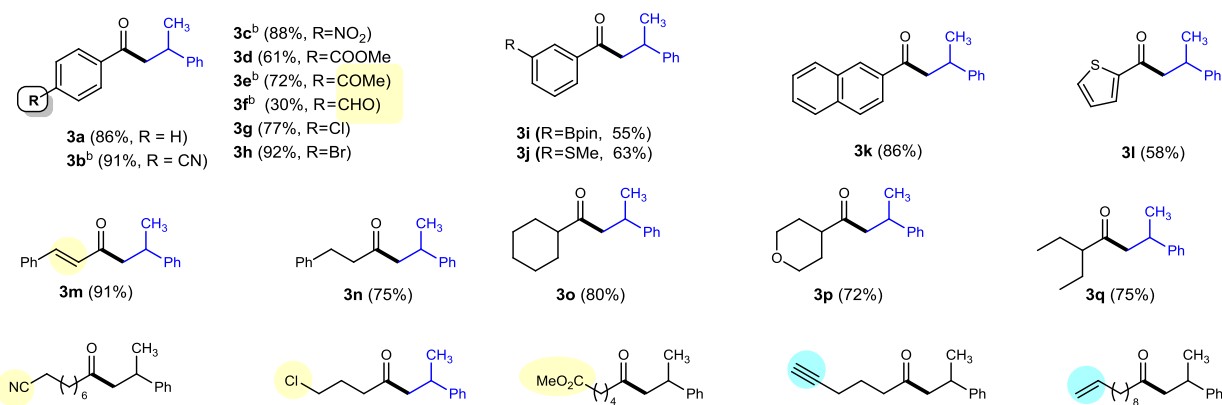

**Scope of amides**

3a (86%, R = H)
3b[b] (91%, R = CN)

3c[b] (88%, R=NO2)
3d (61%, R=COOMe)
3e[b] (72%, R=COMe)
3f[b] (30%, R=CHO)
3g (77%, R=Cl)
3h (92%, R=Br)

3i (R=Bpin, 55%)
3j (R=SMe, 63%)

3k (86%)

3l (58%)

3m (91%)   3n (75%)   3o (80%)   3p (72%)   3q (75%)

3r (74%)   3s (74%)   3t (60%)   3u (50%)   3v (60%)

*Chemoselective for amides in the presence of other carbonyls halides tolerated*

*Orthogonal to metal-catalysed procedures alkenes and alkynes tolerated*

**Scope of alkenes**

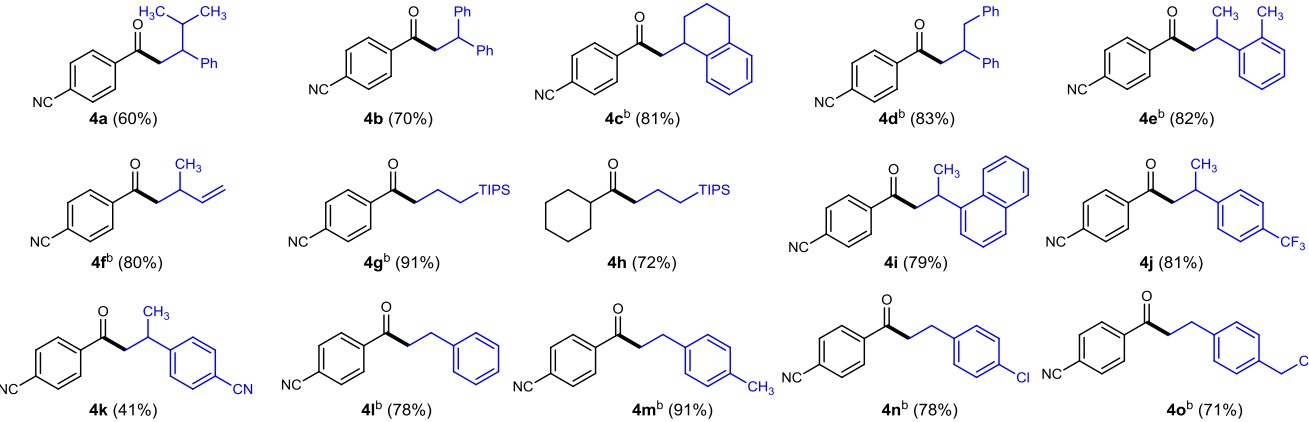

4a (60%)   4b (70%)   4c[b] (81%)   4d[b] (83%)   4e[b] (82%)

4f[b] (80%)   4g[b] (91%)   4h (72%)   4i (79%)   4j (81%)

4k (41%)   4l[b] (78%)   4m[b] (91%)   4n[b] (78%)   4o[b] (71%)

**Late-stage hydroacylation of bioactive substrates**

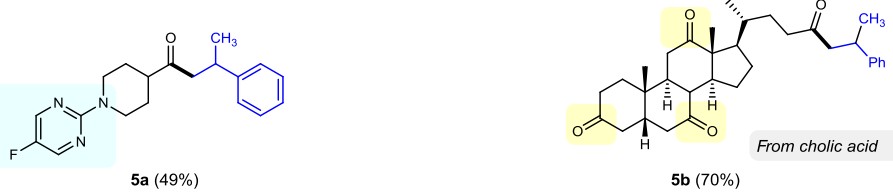

5a (49%)

5b (70%)   *From cholic acid*

[a]Reaction conditions: **1** (0.2 mmol), Tf2O (0.22 mmol), 2-F-Py (0.22 mmol), alkene **2** (0.4 mmol) in DCM (0.1 M) at 0 °C for 2 h and warmed up to room temperature for 12 h under argon; isolated yield after chromatography
[b]CH3CN was used instead of DCM

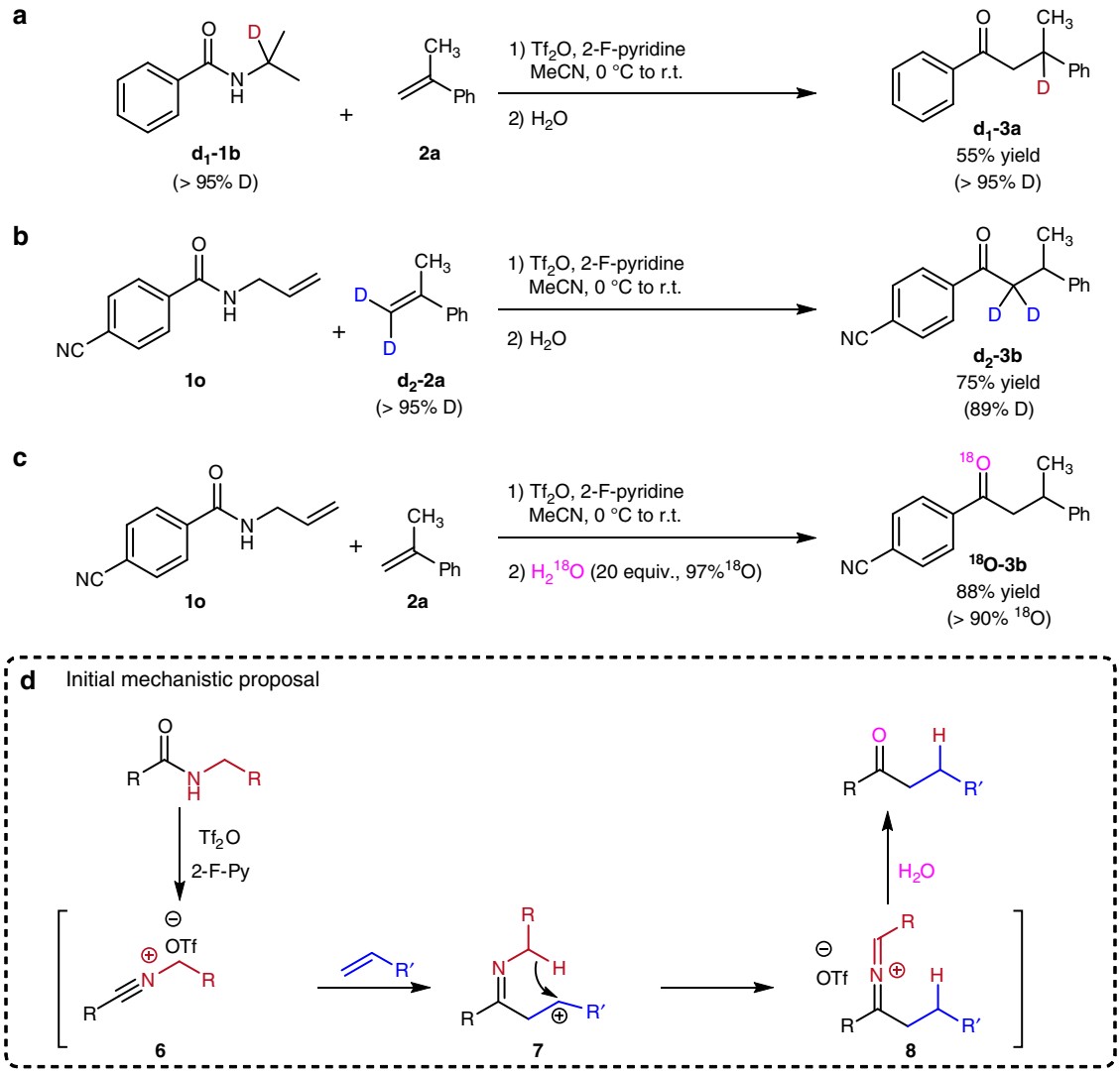

**Fig. 2** Control studies and proposed mechanism. **a** Hydroacylation of **d₁-1b** and **2a** under the optimized condition. **b** Hydroacylation of **1o** and **d₂-2a** under the optimized condition. **c** H₂¹⁸O was used for quenching the reaction. **d** Proposed reaction mechanism

by a $S_N2$-type displacement with a typical trigonal bipyramidal transition state **TS_frag**.

Phenyl and vinyl R-substituents are also compared in Fig. 3. The energy profiles are found to be similar for both substituents, though a slight difference should be mentioned. In the case where R = vinyl, the barrier **6'** → **7** is smaller than the fragmentation barrier ($\Delta\Delta G^{\ddagger} = \Delta G^{\ddagger}(\mathbf{6'} \rightarrow \mathbf{7})$ - $\Delta G^{\ddagger}(\mathbf{6'} \rightarrow \mathbf{frag}) = 24.4 - 25.9 = -1.5$ kcal mol⁻¹), while for R = phenyl the converse occurs: the barrier **6'** → **7** is larger than that for the side-reaction ($\Delta\Delta G^{\ddagger} = \Delta G^{\ddagger}(\mathbf{6'} \rightarrow \mathbf{7})$ - $\Delta G^{\ddagger}(\mathbf{6'} \rightarrow \mathbf{frag}) = 22.0 - 21.5 = 0.5$ kcal mol⁻¹). Overall, this corresponds with the experimental evidence of poor yield in the case of phenyl substituent and is suggestive of a scenario where R = Ph for which the side-reaction becomes more feasible.

## Discussion
The herein presented approach to olefin hydroacylation is a method whereby ketone synthesis can be achieved by the direct, metal-free coupling of secondary amides and alkenes. This obviates the need for transition metal catalysis and proceeds by a new mechanism that neither requires directing

groups nor suffers from deleterious decarbonylation. This work also showcases the potential of carefully designed internal hydride transfer events to provide unique solutions for fundamental, contemporary challenges in organic synthesis.

## Methods
**General procedure for the metal-free hydroacylation of alkenes.** A flame dried Schlenk under argon was charged with the allyl amide **1** (0.2 mmol), and 2-fluoropyridine (0.22 mmol) in 1 mL dry solvent. The mixture was cooled to 0 °C and freshly distilled trifluoromethanesulfonic anhydride (0.22 mmol) was slowly added and stirred for 15 min, then alkene **2** (0.4 mmol) was added, the reaction was stirred for 2 h at 0 °C and warmed up to room temperature. After stirring for 14 h, the reaction was quenched with 10 mL 1 M HCl. The layers were separated and the aqueous layer was extracted with DCM (3 × 10 mL). The combined organic layers were dried on MgSO₄, filtered and evaporated in vacuo. The product **3**, **4** and **5** were purified on column chromatography (Heptane/MTBE = 99:1 to 8:2).

## Data availability
The authors declare that the data supporting the findings of this study, including synthetic procedures, NMR spectra, characterization for all new compounds and further details of computational studies, are available within the article and its supplementary information files, or from the corresponding author upon reasonable request.

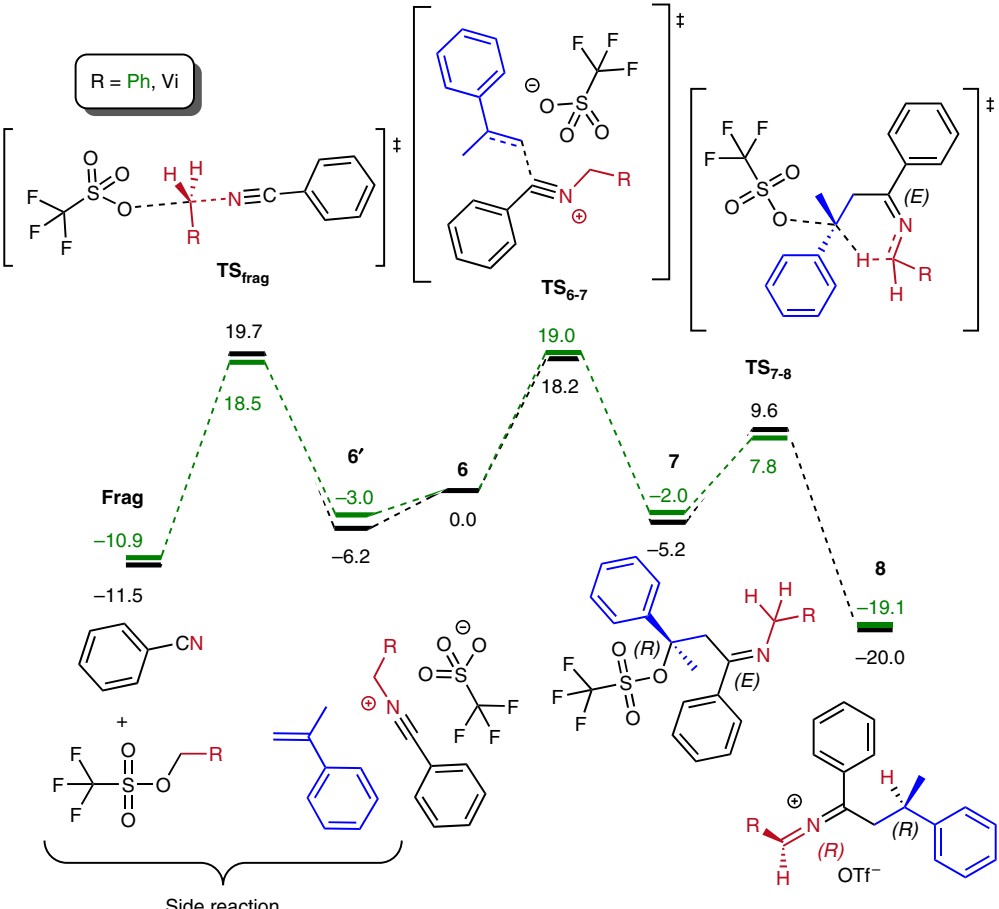

**Fig. 3** Computational results. Computed reaction profile (DLPNO-CCSD(T)/def2-TZVP//B3LYP-D3/def2-SVP, $\Delta G_{298,DCM}$, kcal mol$^{-1}$) for formation of the final intermediate **8** starting from the nitrilium intermediate **6** (reactant complex, taken as a reference 0.0 kcal mol$^{-1}$) and the side-reaction yielding benzonitrile for the vinyl (black) and phenyl (green) substituent R

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

## Acknowledgements

We are grateful to the University of Vienna for the continued support of our research programs. Financial support by Austrian Science Fund (FWF, Grant P30226) and the European Research Council (CoG 682202, VINCAT). Calculations were partially performed at the Vienna Scientific Cluster (VSC).

## Author contributions

J.L. and R.O. performed and analysed the experiments, J.L., R.O., B.M. and N.M. co-wrote the manuscript. B.M. and L.G. performed the computational studies. N.M. directed the project.

## Additional information

**Competing interests:** The authors declare no competing interests.

