## [Peer Review File · Nature Communications]

Reviewers' comments:

Reviewer #1 (Remarks to the Author):

The manuscript reports a redox-neutral hydroacylation of alkenes using secondary amides as coupling partners. Hydroacylation reactions are among the most important methods for efficient construction of C-C bonds. The method advances the concept of metal-free electrophilic activation of amides to the classic hydroacylation chemistry. This is a very interesting paper that I strongly support for publication in *Nat. Commun.* The translation of the redox-neutral coupling of olefins with amides may access new modes of reactivity of the fundamental amide bond and is particularly worthy of publication.

Before publication, the authors should make the following changes:

1. In the introductory paragraphs, they should reference that a Pd-catalyzed synthesis of ketones is now more established than Ni-catalyzed approach. The following reference could be cited: *Acc. Chem. Res.* 2018, 51, 2589-2599. Figure 2b should be revised to include Pd.
2. In the discussion of Table 1 and mechanism, it would be helpful to expand on the high reactivity of allylamides. Have they tried t-Bu amides in this reaction? What is the reason for low reactivity of N-Bn amides? It appears that 1,5-H shift to 8 would be facile.
3. In the discussion of Table 2, overstatements should be avoided. Ketones like 3e and aldehydes like 3f are easily tolerated in metal-catalyzed cross-couplings of amides with olefins. A related method on coupling amides with olefins could be cited: *Angew. Chem. Int. Ed.* 2015, 54, 14518-14522.
4. It would be helpful to mention if tetrasubstituted olefins and simple linear alpha olefins are tolerated.
5. In Figure 2d, olefin could be added over the arrow.
6. A short sentence outlining what happens with the N-substituent could be added in the mechanism.

Reviewer #2 (Remarks to the Author):

This short and ultra-dense contribution reports on redox-neutral synthesis of ketones by coupling of alkenes and amides. The authors themselves suggest this point of view when they write that the hydroacylation of alkenes employs amides in a metal-free regime, proceeding by an entirely new mechanism and offering orthogonal reactivity to the conventional, metal-catalyzed alternatives. With this in mind, this contribution mainly corresponds to new organic synthesis methodology and mechanism. Based on mainly the selective deuterium incorporation, H₂O(18) quenching and NMR analysis, a possible mechanism is available but only assumptions. Altogether, the only partial characterization of the reaction combined with some systematic, but empirical, exploitation of appealing mechanism for high-impact journals (metal-free catalysis for ketone synthesis) are still premature. Therefore, major revision should be made before it becomes suitable for publication.

1. Why the author defines the reaction time as 14 h but not via the TLC? And how does the author monitor the possible mediate/product during such a long period to conduct the mechanistic studies?
2. How does the substituent affect the reaction yield and, probably, the reaction rate? Further studies including detailed computational works on the mechanism should be very necessary to support the author's assumption.
3. Compared with 3a-3e, 3g, 3h, why the obtained yield of 3f is much lower?
4. In Table 2, the scheme, what is the meaning of 0.2M?
5. The deuterium experiment is not sufficient enough, for example KD/KH should be given.

Reviewer #1

1. In the introductory paragraphs, they should reference that a Pd-catalyzed synthesis of ketones is now more established than Ni-catalyzed approach. The following reference could be cited: *Acc. Chem. Res.* 2018, 51, 2589-2599. Figure 2b should be revised to include Pd.

→ We already added the reference in new draft.

2. In the discussion of Table 1 and mechanism, it would be helpful to expand on the high reactivity of allylamides. Have they tried *t*-Bu amides in this reaction? What is the reason for low reactivity of *N*-Bn amides? It appears that 1,5-H shift to 8 would be facile.

→ The presence of an α -H is required for 1,5-H shift and unfortunately *t*-Bu amides don't have one. Based on newly performed DFT calculations, *N*-Bn and *N*-allyl amide show similar barrier for the 1,5-H shift event. However, the *N*-Bn nitrilium species 6 is unstable, and easily undergoes fragmentation to the corresponding nitrile. See Figure 3 and ref. 25 (*Tetrahedron* 71, 3795-380 (2015))

3. In the discussion of Table 2, overstatements should be avoided. Ketones like 3e and aldehydes like 3f are easily tolerated in metal-catalyzed cross-couplings of amides with olefins. A related method on coupling amides with olefins could be cited: *Angew. Chem. Int. Ed.* 2015, 54, 14518-14522.

→ We thank the reviewer for this comment. We have changed this and added the references in the revised version.

4. It would be helpful to mention if tetrasubstituted olefins and simple linear alpha olefins are tolerated.

→ We have included a statement to this end.

5. In Figure 2d, olefin could be added over the arrow.

→ This was added.

6. A short sentence outlining what happens with the *N*-substituent could be added in the mechanism.

→ A sentence was added "Upon hydrolysis, ammonia and an aldehyde are formally excised from 8".

Reviewer #2

1. Why the author defines the reaction time as 14 h but not via the TLC? And how does the author monitor the possible mediate/product during such a long period to conduct the mechanistic studies?

→ Indeed, we monitored the reaction by TLC. We found that in most cases the reactions are completed after 14 h.

2. How does the substituent affect the reaction yield and, probably, the reaction rate? Further studies including detailed computational works on the mechanism should be very necessary to support the author's assumption.

→ We have performed added quantum chemical calculations to get a deeper insight into the reaction mechanism. The calculations and the respective discussion have been added to Figure 3, providing a much more complete panorama. The computational results are consistent with our experimental observations.

3. Compared with 3a-3e, 3g, 3h, why the obtained yield of 3f is much lower?

→ We found that product 3f unfortunately decomposed during purification. However, the crude reaction mixture was quite clean.

4. In Table 2, the scheme, what is the meaning of 0.2M?

→ 0.2 M means that the concentration of substrate **1** was 0.2 mmol/mL. The SI contains detailed information on the experimental procedure.

5. The deuterium experiment is not sufficient enough, for example K_D/K_H should be given

→ With the newly added quantum chemical calculations, a deeper insight into the reaction mechanism is provided, also regarding the relative rates of events.

REVIEWERS' COMMENTS:

Reviewer #1 (Remarks to the Author):

All points have been addressed by the authors. The manuscript is suitable for publication in the present form.

Reviewer #2 (Remarks to the Author):

All the comments raised by the reviewers have been addressed. So the manuscript could be published as is.

REVIEWERS' COMMENTS:

Reviewer #1 (Remarks to the Author):

All points have been addressed by the authors. The manuscript is suitable for publication in the present form.

→**Thanks for Reviewer #1**

Reviewer #2 (Remarks to the Author):

All the comments raised by the reviewers have been addressed. So the manuscript could be published as is.

→**Thanks for Reviewer #2**